# Xenocentrism and Consumer Buying Behavior: A Comparative Analysis of Malaysian vs. Nigerian Consumers

**Vimala Venugopal** [1], **Mário Nuno Mata** [2], **Chiemelie Benneth Iloka** [3,*], **Rui D. Dantas** [2], **Anabela Batista** [2], **João Xavier Rita** [2] **and José Moleiro Martins** [2,4]

1. Marketing Department, Taylor's School of Business and Law, Taylor's University Malaysia, Jalan Taylors, Subang Jaya 47500, Malaysia; Vimala.Venugopal@taylors.edu.my
2. ISCAL-Instituto Superior de Contabilidade e Administração de Lisboa, Instituto Politécnico de Lisboa, Avenida Miguel Bombarda 20, 1069-035 Lisboa, Portugal; mnmata@iscal.ipl.pt (M.N.M.); rmdantas@iscal.ipl.pt (R.D.D.); ambatista@iscal.ipl.pt (A.B.); jmrita@iscal.ipl.pt (J.X.R.); zdmmartins@gmail.com (J.M.M.)
3. Marketing Department, Enugu State University of Science and Technology (ESUT), PMB 01660, Agbani 402004, Nigeria
4. Instituto Universitário de Lisboa (ISCTE-IUL), Business Research Unit (BRU-IUL), 1649-026 Lisboa, Portugal
* Correspondence: iloka.benneth@esut.edu.ng

**Abstract:** Across the globe, xenocentrism has emerged as a critical concept for understanding the behavior of consumers towards the purchase of local and foreign products. In line with this context, and based on samples collected from Malaysia and Nigeria, this study analyzed the direct effects of xenocentrism on the buying behavior of consumers towards imported products with perceived product quality, purchase intention, and product attitude as the variables used to measure consumers' buying behavior. XSCALE was used to measure xenocentrism, and the research model was analyzed with the partial least squares form of structural equation modeling. A total of 400 responses were gathered from Malaysia, whereas 453 responses were gathered from Nigeria. Findings from the investigation show that xenocentrism has a positive influence on perceived product quality (Malaysia = 46.7%, Nigeria = 35%), purchase intention (Malaysia = 46%, Nigeria = 47.3%), and product attitude (Malaysia = 39.2%, Nigeria = 38.4%), Based on these findings, this study concluded that xenocentrism is a valid construct for assessing the purchase behavior of consumers in Malaysia and Nigeria towards foreign products competing in their local markets. On the same note, the findings from this research can be used to develop sustainable marketing strategies suitable for xenocentric consumers across Malaysia and Nigeria (in particular), and the entire developing economies (in general).

**Keywords:** attitude; behavior; consumer; products; purchase intention; xenocentrism; AMOS-SEM; M300; M310

## 1. Introduction

As marketers continue to assess the factors that influence consumers' purchase behavior, one of the reoccurring questions is: why are some consumers content, or perhaps even excited, about the hope of purchasing foreign products, whereas other consumers, on the other hand, are resistant to purchasing foreign products? The answer to this question lies in understanding consumer centrism. Notwithstanding the increased importance of this question, studies (such as Lawrence 2012) point out that markets do not have a solid grasp of the complete array of factors that can influence the opinion consumers have about the appropriateness of buying foreign products. Therefore, it can be said that marketers are ill-equipped to answer such a vital question in the marketplace that is increasingly globalizing (Lawrence 2012).

For those that prefer foreign products, this behavior is termed xenocentrism. Xenocentrism is defined as the disproportionate and unreflected attention that consumers accord

foreign products, ideas, or values (Kent and Burnight 1951). Although there is a defined and accepted concept of what consumer xenocentrism is all about, it is generally viewed as the opposite of ethnocentrism (Kent and Burnight 1951; Prince et al. 2016), but that is not the reality of what it entails. In turn, this has created a problem when it comes to interpreting the scholarly works designed to assess xenocentrism (Kent and Burnight 1951; Prince et al. 2016). Furthermore, studies have not looked at a comparative analysis of xenocentrism across cultures in order to assess the supposedly prerogative view of generalization of cultures (Kent and Burnight 1951; Prince et al. 2016). This forms the foundation of this present study, which seeks to compare the influence of xenocentrism on two cultures (Nigeria and Malaysia).

This study aims to comparatively analyze the influence of xenocentrism in Malaysia and Nigeria, as it relates to consumers' purchase intention. The elements of purchase intention adopted for the study are: perceived product quality, purchase intention, and product attitude. Overall, the essence of the study is to: (a) fill the gap and need in research as it was found that much has not been carried out on the two nations and there is no existing comparative study between them; and (b) to expand existing understandings, models, and theories related to xenocentrism and consumers' purchase intention. By attaining this objective, the researchers will lay the foundation for new views while solidifying the existing ones. This paper is divided into four main sections. The first is the introduction which lays the foundation for the entire study, followed by the literature review which reviews extant and current studies related to the work, and research methods containing the approaches used for data gathering and analysis. The final sections are the Discussion of the findings and the Conclusion based on the entire research.

## 2. Literature Review

As a result of the globalization of the marketplace, it is now customary for consumers to make purchase decisions related to buying either domestic or foreign manufactured products. However, although this globalization is seen to be continuously increasing, as marketers, one is never too sure if the consumers are globalizing themselves along with the marketplace (Cleveland et al. 2009). Thus, there are different views (mostly disagreement) from theorists that have looked at whether consumers are becoming globalized and whether they are more likely to purchase foreign goods (Lawrence 2012). From one side of the argument, the view that stands is that different market factors (such as capitalism, communication, global transport, advertising and marketing, and cosmopolitanism at the transnational levels) are interacting with each other, yielding the dissolution of the global boundaries across different national economies and cultures (Ger 1999); and many scholars (e.g., Alden et al. 1999; Firat 1995; Hannerz 1990; Ter Hofstede et al. 1990) assert that this has brought about accelerated emergence of a homogenous culture on global consumption (Cleveland et al. 2009, p. 116).

The other side of the argument has theorists saying that, notwithstanding the deluge of international influences that the average consumers experience, the consumption patterns of consumers are a product of their local cultures. De Mooij (2004) further argued that the strong push towards globalization has resulted in consumers being increasingly persistent in their attachment towards local cultures. The impact is that the appeal of foreign-made products is thwarted. The unfortunate thing is that there exist limited studies in this area, making it difficult to determine the side of the argument with the right point-of-view. One could argue that both sides are correct, making it possible for customers to be segmented based on this divide. In any case, to identify these varying segments of consumers based on their willingness to buy foreign-made products, the first step will be to measure the necessary constructs (Cleveland et al. 2009). Shreds of evidence also show that till present, there are limited repertoires of such scales in the stream of marketing tools, and those available are not adequate when it comes to sufficiently measuring all the latent variables capable of influence the decision of customers to purchase foreign goods, another gap that exist in this literature.

### 2.1. Conceptualization of Xenocentrism

Xenocentrism is conceptualized to entail consumers naively devaluing their cultural values, and in the process, overrating the values of foreign cultures. The term (xenocentrism) was first coined in the early 1950s by Kent and Burnight (1951), who considered it to be "a view of things where the center of attraction is on a group other than one's group, and all other groups, including the one the person is affiliated to, are rated based on the groups (external) that take center stage." When people show these features, they are bound to develop a preference for elements and ideas from the cultural communities different from their own native culture.

In more recent times, studies have viewed xenocentrism as the disproportionate and unreflected focus that one has towards others (Stier 2010), which results in a high preference for imported commodities or ideas. Essentially, consumer xenocentrism leads to people voluntarily paying more for imported commodities than they would typically be willing to pay for the locally manufactured alternatives (Stier 2010).

Put simply, xenocentric individuals have a preference for society (in terms of cultures, values, norms, and so on) other than their own, rating and scaling everything by referencing the society instead of their own (Kent and Burnight 1951). Different reasons exist for which a consumer might feel attached to other societies. The most prevalent reason for a high level of xenocentrism, considering the increasing homogeneity of the global market, is consumers being firmly attached to their ancestral lands. Historically, it has been demonstrated that xenocentrism, from a cultural perspective, is a common phenomenon, even in post-colonial societies, with documented evidence showing that people consider foreign culture superior to all others, including their own culture (Prince et al. 2016). Good examples abound in today's world, where countries such as the United States, United Kingdom, and Germany are examples of global superpowers that export "super products" backed by media and saturated product marketing. This is the main reason why products manufactured in these cultures might seem to be the preferred option, especially to customers in the developing and sectional economies, and this helps in fermenting xenocentrism.

Even in academic research, this view is also helped when one considers the most common paradox: the assumptions and theoretical frameworks conceptualized in these developed economies can be applied across the broad spectrum of culture, religions, economies, and ethnicities (De Mooij 2013). However, consumers are also known to evaluate the psychological and tangible features of a product based on their motivations, native societal norms, and customs. Therefore, the most crucial factors shaping the way consumers view a foreign brand are the political, social, and economic factors aligned with cultural values (Chapa et al. 2006). As such, the phenomenon of xenocentrism might not be explained or perceived in similar ways and might be nonexistent in countries where the social determinants that ferment its essence and effect are not experienced. Within the dimension of theories related to comparison and social power, it is asserted that foreign products are accorded more positive evaluation in the developing economies (e.g., Batra et al. 2000; Laroche et al. 2005) as the consumption of these foreign goods is used to illustrate the social status of the people (e.g., Steenkamp 2001; Yeniyurt and Townsend 2003; Craigh et al. 2005).

### 2.2. Perceived Product Quality

The perceived quality of a given product emerges as a result of the convention between the seller and buyer, and it is a multidimensional entity that has been conceived differently by various research schools of thought (Stylidis et al. 2019). Within the context of marketing, perceived quality is used to reference the consumer's general interpretation of the quality of a given product or service, which concerns the reasons that surround the consumer's decision to choose a given product instead of an alternative brand (Aaker 1997). On the essence of perceived quality, Zeithaml (1988) pointed out that perceived quality is the consumer's judgment about the overall essence or superiority of a given product against

other alternatives. As such, such an assessment will be based on various product-based features.

Various researchers have demonstrated the idea of perceived product quality (PPQ). In line with the views of Aaker and Joachimsthaler (2000), product quality represents a form of connection that influences how consumers associate a brand in most contexts, and empirical works support its influence on profitability. Additionally, product quality means the features of a product that satisfies consumers' need and could cover different elements that can enhance the performance of a given product (Dunk 2002). Therefore, measuring consumers' requirements is a prerequisite for developing product quality (Rao et al. 1997). This is because before arriving at a purchase decision, consumers have different views about the style, quality, and price of a given product. Product features increase or decrease customers' purchase intention. That is why when the quality of a given product is high, there will be an increase in the purchase intention of customers.

Similarly, xenocentrism has been conceptualized as a representation of social identity. Some authors have developed proposals on how social identity influences consumers' perception about a product (Addis and Grünhagen 2014; Rondán-Cataluña et al. 2015; Badaoui et al. 2018); with a focus on how they perceive the quality of a product (Lantz and Loeb 1998). Therefore, the following hypothesis is proposed.

**Hypothesis 1 (H1).** *Xenocentric consumers will perceive foreign brands as having higher quality than locally manufactured ones, leading to increased purchase intention towards such brands.*

### 2.3. Product Attitude

Following the work of Fishbein and Ajzen (2011), attitude is defined as the aggregated evaluation of feelings someone has about the excellent or harmful nature of a given objective or concept. Based on the work of Ajzen and Fishbein (2005), attitude features two dimensions: (1) attitude attributed to concrete actions, such as making use of social networking sites for funding a project; (2) the attitude attributed towards objects, such as issues, products, or organization. Based on this classification, it can be envisaged that product attitude (PA) directly influences the purchase intention of a given product. The reason being that product attitude compliments the determinants of purchase action, together with effects on vital views that people have towards the purchase and the facilities delivered for such action. Therefore, the purchase intention towards a given product grows when, in addition to the demand being made for the said product, the consumer has a favorable attitude towards the product (Saleem et al. 2015).

Authors have pointed out that consumers develop a more favorable intention to purchase a product when they have a better attitude towards the product (Babin and Babin 2001; Lee and Lee 2009). Additionally, in the present study, xenocentrism is seen as representing social categories at the national level. It is shown in the literature that social variables (such as subjective norms and status) can influence the attitude that consumers have toads purchasing products from different categories (Irianto 2015; Ajitha and Sivakumar 2017). On that note, it is hypothesized that:

**Hypothesis 2 (H2).** *Xenocentric consumers will show a more positive attitude towards foreign brands against locally manufactured ones, leading to increased purchase intention towards such brands.*

### 2.4. Purchase Intention

Several studies have been conducted to assess the influence of xenocentrism on consumer's purchase behavior. Lawrence (2012) looked at consumer xenocentrism, and consumer cosmopolitanism focused on the development and validation of scales of constructs that influence consumers' attitudes towards the consumption of foreign products. In this research, the author created two new scales that can be used to measure xenocentrism

and cosmopolitanism constructs within the context of consumers. Balabanis and Diaman­topoulos (2016) focused on a system justification perfective of consumer xenocentrism as a determinant of preference for foreign products. They pointed out that the decision to purchase foreign products is mainly dependent on the consumer's predisposition, and the dominant construct that explains such behavior is ethnocentrism, anchored on the social identity theory. However, such a view ignores the fact that certain consumers are attracted to the "foreignness of products." Thus, their research was based on system jus­tification theory to conceptualize and offer empirical tests about consumer xenocentrism constructs, intending to explain consumers' attraction towards foreign products. Survey data were gathered from five complementary studies. The researchers developed a new scale (C-XENSCALE) for measuring xenocentric tendencies of consumers, validating the broad shreds of evidence that xenocentrism can be used to explain consumers' preference for foreign products.

Diamantopoulos et al. (2019) focused on modeling the role of consumer xenocentrism in impacting preferences for domestic and foreign brands, based on mediation analysis. The main highlight of their work is that there are limited empirical works on consumer xenocentrism. Their study was based on system justification theory with a sample of 262 Russian consumers. Findings from the study showed that a serial mediation model, with product-country image and brand attitudes serving as the intervening variables, effectively describes the route through which xenocentrism in consumers predict: (a) positive influence on their intentions to purchase genuine foreign brands, and (b) negative influence on their intention to buy domestic brands. In the work of Makri et al. (2019), the focus was on global-local consumer identities as drivers of the usage of global digital brands. A cross-country survey was developed in a developed (Austria) and developing (Thailand) economy, with a total of 425 data points gathered from young adults and analyzed with SEM technique to test a set of hypotheses. Findings show that in Thailand, the users with a global identity enjoy participating in global SNS more than their counterparts in Austria. On the same note, consumers that exhibit local identity in Thailand were found to demonstrate less pleasure when participating in global SNS than their counterparts in Austria, and as such, are less inclined to use global SNS.

In a more recent study, Camacho et al. (2020) looked at the influence of xenocentrism on the purchase intention of consumers and focused on the mediating role of product attitudes. The author acknowledged that xenocentrism has emerged as the global construct for understanding consumer behavior concerning local and foreign products. The study adopted a sample of Colombian consumers to assess the direct and indirect effect of xenocentrism on consumers' purchase intention towards imported products, through perceived product quality and product attitude. Xenocentrism was measured with XSCALE and analyzed with the partial least squares SEM. Findings show that xenocentrism leads to a positive direct effect on purchase intentions, perceived product quality, and product attitude. It was also demonstrated that product attitude partially mediates the effect that xenocentrism has on purchase intentions. From the above empirical review, the reoccurring limitation is that attempts have not been made to comparatively assess the influence of xenocentrism on the purchase intention of consumers across different markets. Even in the work of Makri et al. (2019), the focus was on identity, not the purchase of a product. Therefore, this present study aims to fill this gap. Additionally, there are limited empirical works on xenocentrism, with this research designed to add to the existing portfolio of works to broaden the understanding of xenocentrism and its constructs.

**Hypothesis 3 (H3).** *Xenocentric consumers will prefer foreign brands more than a locally manu­factured one.*

## 3. Materials and Method

From Figure 1, the objectives of this study are illustrated in order to assess the effect of xenocentrism on consumers' behaviour (in relation to perceived product quality, purchase

intention, and product attitude). Questionnaires were adopted as the instrument of data gathering, and the design was based on scales developed in past studies. To measure xenocentrism, the XSCALE, as proposed by Rojas-Méndez and Chapa (2019), was adopted; the scale for perceived product quality was adopted from (Chung and Pysarchik 2000); the scale for product attitude was adopted from (Kim and Pysarchik 2000); the scale used to measure purchase intention was adopted from (MacKenzie et al. 1986). Five items were loaded for xenocentrism, such as "I prefer foreign products more than locally manufactured products"; both perceived product quality and product attitude had four items each, such as "imported products have higher quality", and "it is always a good idea to purchase imported products"; and lastly, purchase intention also has five items, such as "I will likely purchase an imported product before this year ends." A 5-point Liker rating was used as the response format, where answers ranged from "strongly disagree" (1) to "strongly agree" (5).

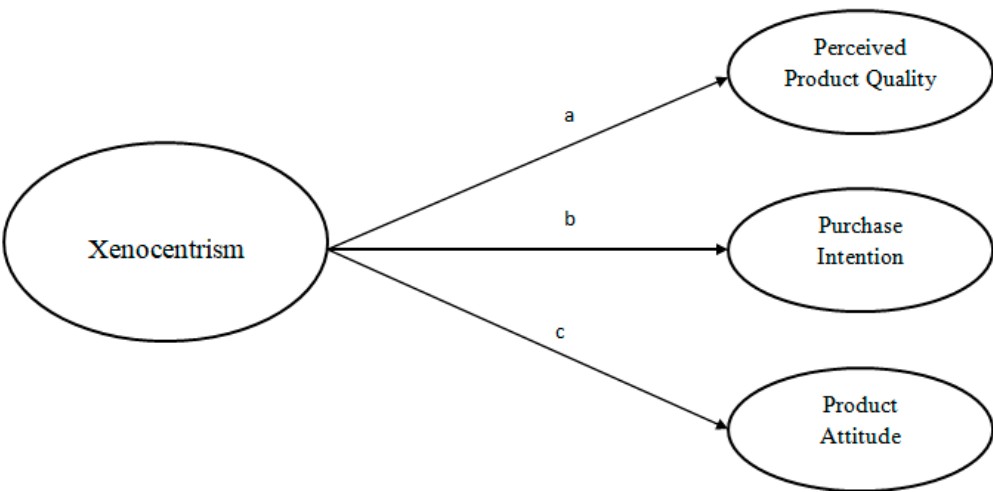

**Figure 1.** Research Model. HI = a, H2 = c, H3 = b.

As a result of the COVID-19 pandemic, the researchers aborted earlier planned face-to-face sampling and adopted an online survey (through GoogleForms). A total of 852 responses (400 from Malaysia and 453 from Nigeria) were gathered. All questions were marked as mandatory, making it impossible to submit an incomplete survey, and eventually, the survey has 100% response rate. The responses were downloaded via XLS file and later transferred (after formatting) to IBM's SEM software for further analysis.

For the questionnaire, the 852 responses gathered were valid. This is because it is impossible to submit the questionnaire without answering all of the questions as they were all marked as mandatory. In terms of gender (see Table 1), the majority (69% for Malaysia and 56.1% for Nigeria) reported being females, followed by their male counterparts (40% for Malaysian and 43.9% for Nigeria). For Malaysia, the majority of the respondents were aged 21–30 years old (29%), followed by 31–40 years old (24%), 18–20 years old (10%), 41–50 years old (13%), whereas 17 years or younger, 51–60 years old, and 60 or above all had less than 10% of the response, respectively. Similar findings were also reported in Nigeria, where the majority of respondents were aged 21–30 years (27.8%), followed by 31–40 years old (26.5%), 18–20 years old (16.8%), and 41–50 years old (14.3); those aged 17 or younger, 51–60 and 60 or above all reported less than 10% of the total response, respectively. The educational level of the respondents was also assessed with respondents from Malaysia reporting that the majority of the respondents have a degree (39%), followed by a certificate/diploma (26%), high school certificate (14%), whereas those with less than high school certificate, master's degree, and doctorate all accounted for less than 10% of the total response, respectively. Similar findings abound in the case of Nigeria where the majority of the respondents have degree certificate (43%), followed by those with certificate/diploma (23.4%), high school

(12.4%), and those with less than a high school certificate, master's degree, and doctorate all accounting for less than 10% of the total response, respectively. The final question from the demographic variable looked at how often respondents purchase foreign products, and responses from Malaysia show that majority buy foreign products at least once per week (42%), followed by at least once per month (38%), and seldom and at least once per year both having 10%, respectively. For Nigeria, the majority buy foreign products at least once per month (41.9%), unlike once per week recorded in Malaysia, followed by at least once per week (38.4%), at least once per year (10.8%), and finally, seldom (8.8%).

**Table 1.** Demographics of the respondents.

| Demographic Variables | Malaysia | | Nigeria | |
|---|---|---|---|---|
| **Gender** | **Frequency** | **Percentage** | **Frequency** | **Percentage** |
| Female | 240 | 60.0 | 254 | 56.1 |
| Male | 160 | 40.0 | 199 | 43.9 |
| **Age** | | | | |
| 17 or younger | 32 | 8.0 | 32 | 7.1 |
| 18–20 | 76 | 19.0 | 76 | 16.8 |
| 21–30 | 116 | 29.0 | 126 | 27.8 |
| 31–40 | 96 | 24.0 | 120 | 26.5 |
| 41–50 | 52 | 13.0 | 65 | 14.3 |
| 51–60 | 12 | 3.0 | 15 | 3.3 |
| 60 or above | 16 | 4.0 | 19 | 4.2 |
| **Education** | | | | |
| Less than high school | 32 | 8.0 | 32 | 7.1 |
| High school | 56 | 14.0 | 56 | 12.4 |
| Certificate/Diploma | 104 | 26.0 | 106 | 23.4 |
| Degree | 156 | 39.0 | 195 | 43.0 |
| Masters | 36 | 9.0 | 45 | 9.9 |
| Doctorate | 16 | 4.0 | 19 | 4.2 |
| **How often do you buy foreign products** | | | | |
| Seldom | 40 | 10.0 | 40 | 8.8 |
| At least once per week | 168 | 42.0 | 174 | 38.4 |
| At least once per month | 152 | 38.0 | 190 | 41.9 |
| At least once per year | 40 | 10.0 | 49 | 10.8 |

To explore the proposed relationships developed in the research model, the researchers employed the partial least squares (PLS) technique. The PLS technique is highly recognized in the marketing discipline (Hair et al. 2012) and considered crucial when it comes to testing the relationship of proposed models (Cepeda et al. 2017). In line with the guidelines offered by (Hair et al. 2017), an evaluation of the measurement model and structural model was undertaken. The Smart PLS 3.0 software was utilized for the said analysis (Nitzl et al. 2016).

## 4. Results

### 4.1. Assessing the Measurement Model

In PLS, the initial evaluation test is to assess the overall goodness of fit of a model. For this study (as shown in Table 2), the overall fit of the model is assessed with adjustment indexes, covering the standardized root mean square residual (SRMR) for the model as estimated. For the SMRM, values of 0.10 were obtained, and these imply that the overall fitness of the model is moderately supported.

**Table 2.** The goodness-of-fit model.

| Fit Criteria | Value | |
|---|---|---|
| | **Malaysia** | **Nigeria** |
| SRMR | 0.100 | 0.100 |
| dULS | 0.899 | 0.903 |
| dG | 0.280 | 0.279 |

Note: All the values of the test (standard root mean square residual (SRMR), unweighted least squares discrepancy (dULS), and geodesic discrepancy (dG)) are under the bootstrap-based 99% percentile.

To assess the appropriateness of the measures, the reliability of both the individual variables and measures were evaluated, together with the convergent and discriminant validity of all the variables loaded in the model. For the individual variables, the reliability was evaluated by assessing the individual loads of the measurement with their variables as contained in the model. The findings show that all the values of the loads associated with the model have values >0.5, with most of them greater than 0.7, above the least acceptable values of 0.5, as featured in the literature. In this analysis, Xenocentrism was considered a hierarchical component with two dimensions of the reflective–reflective form, with the calculation based on guidelines provided by (Hair et al. 2017). The cross-loadings of the variables are provided in Table 3.

**Table 3.** Discriminant validity–cross loadings.

| Indicator | PPQ | | PI | | PA | | XEN | |
|---|---|---|---|---|---|---|---|---|
| | **Malaysia** | **Nigeria** | **Malaysia** | **Nigeria** | **Malaysia** | **Nigeria** | **Malaysia** | **Nigeria** |
| PPQ1 | **0.788** | **0.690** | 0.430 | 0.439 | 0.343 | 0.464 | 0.411 | 0.328 |
| PPQ2 | **0.778** | **0.790** | 0.202 | 0.484 | 0.428 | 0.438 | 0.417 | 0.494 |
| PPQ3 | **0.732** | **0.754** | 0.448 | 0.457 | 0.345 | 0.472 | 0.492 | 0.327 |
| PPQ4 | **0.779** | **0.701** | 0.315 | 0.334 | 0.381 | 0.345 | 0.436 | 0.478 |
| PI1 | 0.420 | 0.411 | **0.738** | **0.790** | 0.346 | 0.439 | 0.344 | 0.354 |
| PI2 | 0.236 | 0.448 | **0.770** | **0.713** | 0.428 | 0.460 | 0.420 | 0.392 |
| PI3 | 0.435 | 0.467 | **0.532** | **0.745** | 0.464 | 0.426 | 0.307 | 0.338 |
| PI4 | 0.313 | 0.374 | **0.669** | **0.601** | 0.353 | 0.372 | 0.451 | 0.317 |
| PA1 | 0.337 | 0.432 | 0.371 | 0.421 | **0.504** | **0.711** | 0.379 | 0.362 |
| PA2 | 0.419 | 0.416 | 0.444 | 0.412 | **0.714** | **0.728** | 0.343 | 0.397 |
| PA3 | 0.390 | 0.472 | 0.480 | 0.499 | **0.767** | **0.672** | 0.352 | 0.458 |
| PA4 | 0.341 | 0.300 | 0.318 | 0.348 | **0.782** | **0.771** | 0.411 | 0.466 |
| XEN1 | 0.445 | 0.318 | 0.387 | 0.349 | 0.346 | 0.473 | **0.589** | **0.799** |
| XEN2 | 0.459 | 0.492 | 0.453 | 0.347 | 0.401 | 0.482 | **0.678** | **0.781** |

Notes: Values in bold represent loadings for each item that are >0.5 (recommended value).

Composite reliability (CR) was adopted as the reliability index for the variables loaded in this study. For this index, the value obtained varied between 0.801 and 0.846, and they are all above the recommended (cut-off) value 0.7. The average variance extracted (AVE) was computed as the basis for assessing the convergent validity for each variable. The values obtained for all the variables are above 0.5, indicating that the entire variables have convergent validity. The coefficients of the variables are documented in Table 4.

**Table 4.** Coefficient of variables.

| Latent Variable | CR | | AVE | |
| :---: | :---: | :---: | :---: | :---: |
| | **Malaysia** | **Nigeria** | **Malaysia** | **Nigeria** |
| PPQ | 0.802 | 0.845 | 0.577 | 0.515 |
| PI | 0.837 | 0.801 | 0.503 | 0.548 |
| PA | 0.819 | 0.838 | 0.594 | 0.532 |
| XEN | 0.846 | 0.899 | 0.693 | 0.519 |

Notes: Composite reliability (CR); average variance extracted (AVE).

For the discriminant validity of the model, the Fornell–Larcker test was employed, assessing whether the square root of the AVE obtained from each construct was above the correlation coefficients obtained for other constructs. On the same note, values obtained from the heterotrait-monotrait ratio indicate that the scores are below the threshold of 0.9. Therefore, these results show that there is discriminant validity between variables loaded in the model, and they are documented in Tables 5 and 6, respectively.

**Table 5.** Discriminant validity—Fornell–Larcker criterion.

| | PPQ | | PI | | PA | | XEN | |
| :---: | :---: | :---: | :---: | :---: | :---: | :---: | :---: | :---: |
| | **Malaysia** | **Nigeria** | **Malaysia** | **Nigeria** | **Malaysia** | **Nigeria** | **Malaysia** | **Nigeria** |
| PPQ | 0.795 | 0.783 | 0.442 | 0.487 | 0.530 | 0.454 | 0.488 | 0.593 |
| PI | 0.480 | 0.435 | 0.741 | 0.724 | 0.454 | 0.533 | 0.531 | 0.599 |
| PA | 0.536 | 0.498 | 0.543 | 0.558 | 0.763 | 0.750 | 0.527 | 0.404 |
| XEN | 0.483 | 0.532 | 0.505 | 0.627 | 0.540 | 0.491 | 0.777 | 0.738 |

Notes: Shown on the diagonal are the square roots of average variances extracted.

**Table 6.** Discriminant validity—heterotrait-monotrait ratio.

| | PPQ | | PI | | PA | | XEN | |
| :---: | :---: | :---: | :---: | :---: | :---: | :---: | :---: | :---: |
| | **Malaysia** | **Nigeria** | **Malaysia** | **Nigeria** | **Malaysia** | **Nigeria** | **Malaysia** | **Nigeria** |
| PPQ | | | | | | | | |
| PI | 0.756 | 0.683 | | | | | | |
| PA | 0.817 | 0.779 | 0.540 | 0.638 | | | | |
| XEN | 0.788 | 0.801 | 0.799 | 0.758 | 0.742 | 0.689 | | |

### 4.2. Summary of PLS Results

Camacho et al. (2020) and Hair et al. (2017) stated that in determining the influence of a given variable on the other, a significance value above 80% indicates strong influence, and a value between 20% and 80% indicate partial influence. In comparison, a value of less than 20% indicates no influence. Findings from Figure 2 support HI (Malaysia = 46.7%, Nigeria = 35%), H2 (Malaysia = 39.2%, Nigeria = 38.4%), and H3 (Malaysia = 46%, Nigeria = 47.3%). The implication is that xenocentrism has a significant positive influence on perceived product quality, purchase intention, and product attitude of consumers in both Malaysia and Nigeria.

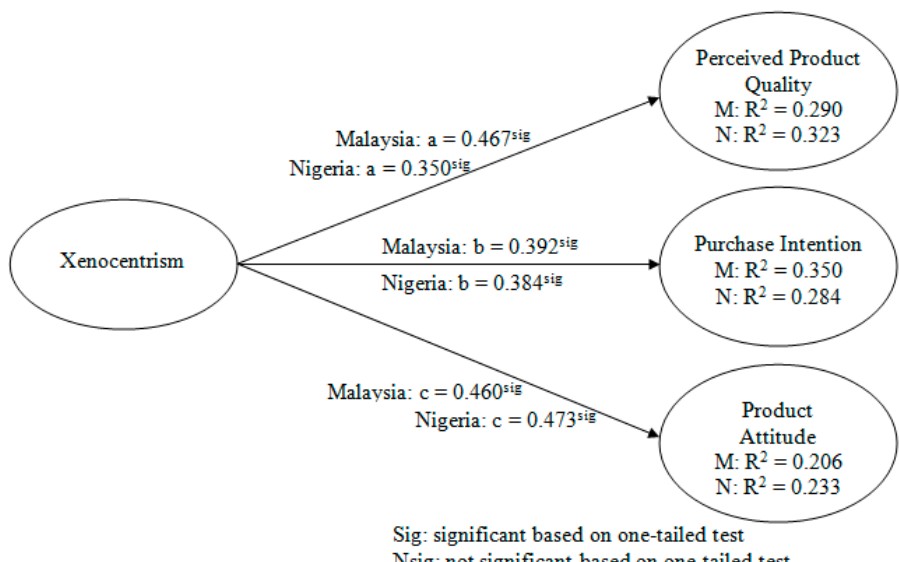

Sig: significant based on one-tailed test
Nsig: not significant based on one-tailed test

**Figure 2.** Summary of PLS Result. HI = b H2 = a H3 = c.

## 5. Discussion

Findings from this study support of the model developed, providing practical and theoretical implications for further related studies on consumer behavior, with a special focus on the influence of xenocentrism on consumer behavior. The first of these implications is that the findings provided in this study validate the XSCALE developed by Rojas-Méndez and Chapa (2019) as a useful construct in measuring consumer xenocentrism. The results show that xenocentrism has a direct positive influence on perceived product quality, in line with the work of Belk (1988) that focused on the predisposition of customers toward local products and Balabanis and Diamantopoulos (2016) that focused on the conviction of the mediocrity of local items. A similar influence was found in the case of purchase intention and product attitude, in line with the finding from Batra et al. (2000). These findings play a pivotal role in firms that are into international business since the appreciation of foreign products might pave the path for intensification of their commercialization process. On the same note, xenocentrism has emerged as a vital concept within the dimension of consumer behavior, primarily when consumers attribute a higher quality to imported products in comparison with the locally manufactured ones. The results indicate that xenocentric consumers have higher expectations about quality. This is in line with the work of Zeithaml (1988) and Camacho et al. (2020), where xenocentric individuals were found to have higher expectations about factors such as quality of products, price, durability, and environmental protection, and these factors yield significant influence on their overall assessment of a given product.

## 6. Conclusions

There are three limitations to this research, which could pave the right path for further related studies. The first is that this study did not put product categories and industries into consideration. Assessing and comparing the influence of xenocentrism among various types of products, services, and industries would yield useful results that international marketers and business professionals can be used to develop the right strategies for given industries and product categories. Secondly, although two countries were compared, it is impossible to generalize these results across developing economies. Therefore, new studies should seek to compare three or more countries based on categories of development. In any case, it is still important to point out that the use of a non-representative sample, as is the case of this study, is still related to the fundamental issues of research concerning xenocentrism. Finally, electronic data were gathered, and this should be taken with caution because those that have access to the internet might be considered more inclined towards

foreign products than those that do not have such access. Therefore, the result might show a stronger relationship between the variables studied than what could be observed in offline research. Further research should see to apply a face-to-face survey or a combination of online and offline surveys.

This research can be concluded based on the idea that xenocentrism explains the purchase intention of consumers towards foreign products, with xenocentrism influencing perceived product quality, purchase intention, and product attitude. Therefore, xenocentrism can be used to explain the purchase behavior of consumers towards foreign brands in Malaysia and Nigeria, with a particular focus on how these consumers perceive the quality of products, their attitude towards products, and the decision to purchase imported products competing in their respective local markets. These findings can also be applied to developing nations as both countries fall within developing economies.

**Author Contributions:** Conceptualization, V.V. and C.B.I.; methodology, M.N.M. and C.B.I.; software, R.D.D.; validation, A.B., J.X.R., and J.M.M.; formal analysis, R.D.D.; investigation, V.V., C.B.I., and M.N.M.; resources, J.X.R., and J.M.M.; data curation, V.V. and C.B.I.; writing—original draft preparation, R.D.D.; writing—review and editing, C.B.I. and M.N.M.; visualization, M.N.M.; supervision, M.N.M.; project administration, A.B. All authors have read and agreed to the published version of the manuscript.

**Funding:** This research received no external funding.

**Institutional Review Board Statement:** Not applicable.

**Informed Consent Statement:** Not applicable.

**Data Availability Statement:** The data for this study is available upon request. To access the data, please contact the corresponding author iloka.benneth@esut.edu.ng.

**Conflicts of Interest:** The authors declare no conflict of interest.

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
