# Peer review of "Xenocentrism and Consumer Buying Behavior: A Comparative Analysis of Malaysian vs. Nigerian Consumers"

_jrfm, doi:10.3390/jrfm15020058_

Round 1

Reviewer 1 Report

Dear Authors, 

Thank you for the opportunity to read and review your manuscript submitted to the Journal of Risk and Financial Management. After reading the manuscript I can congratulate you on the comprehensive research on xenocentrism and consumer buying behavior. It was a pleasure to read your manuscript. Even though you did a huge job, I have some recommendations and comments on how to strengthen the manuscript before publication: 

  1. The introduction needs to be expanded. You should more clearly and thoroughly state the motives for the research, its contribution to the current scholarship. 
  2. All your hypotheses are numbered as H3. Please check this issue. 
  3. The title of 2.4 points empirical review. It is a part of the literature review. Therefore, my suggestion would be to move relevant texts for the justification of your hypotheses rather than presenting them in a separate chapter that does not clearly lead to your hypotheses. 
  4. Please check all the titles of the sections with the requirements of the Journal. 
  5. The Discussion is missing. Please, prepare this part. 

Once again, thank you for the opportunity, and wish you good luck in strengthening the manuscript. 

Author Response

  1. The introduction needs to be expanded. You should more clearly and thoroughly state the motives for the research, its contribution to the current scholarship. 

Suggestion adopted: Expanded to include motive and contribution

  1. All your hypotheses are numbered as H3. Please check this issue. 

Suggestion adopted: Properly numbered the hypotheses

  1. The title of 2.4 points empirical review. It is a part of the literature review. Therefore, my suggestion would be to move relevant texts for the justification of your hypotheses rather than presenting them in a separate chapter that does not clearly lead to your hypotheses. 

Suggestion adopted: Integrated into the literature review

  1. Please check all the titles of the sections with the requirements of the Journal. 

Suggestion adopted: arranged based on journal’s requirement

  1. The Discussion is missing. Please, prepare this part

Suggestion adopted: Discussion is added to the work

Reviewer 2 Report

Please, use the format of references of the journal.

In Abstract
1. Please, induce the number of participants in both countries.
2. You mention direct and positive in the same statement. But direct = positive. and reverse = negative. Please fix it. 
3. Please, mention the beta obtained between the variables.
4. Include in keywords: purchase intention, PLS-SEM

Introduction
1. In the final must be included the sections of the article.

Theoretical background
1.  It mentions H3 but H1 and H2 are missed.

Research objective
1. The research model doesn't follow the TPB proposed by Ajzen. The sequences must be Xenocentrism --> Attitude --> Intention to purchase

It means a recalculation of the variables and a re-write of the rest of the article.

Author Response

In Abstract
1. Please, induce the number of participants in both countries.

Done

  1. You mention direct and positive in the same statement. But direct = positive. and reverse = negative. Please fix it. 

Done

  1. Please, mention the beta obtained between the variables.

Included, in the form of percentage of influence obtained through path analysis

  1. Include in keywords: purchase intention, PLS-SEM

Done

Introduction
1. In the final must be included the sections of the article.

Done

Theoretical background
1.  It mentions H3 but H1 and H2 are missed.

Done

Research objective
1. The research model doesn't follow the TPB proposed by Ajzen. The sequences must be Xenocentrism --> Attitude --> Intention to purchase It means a recalculation of the variables and a re-write of the rest of the article.

This article was based on a modified model (supported by Ajzen’s proposed TPB) and Xenocentrism was measured by XSCALE as proposed by Rojas-Méndez and Chapa (2019). Thus, there is no need to follow the exact sequence of Ajzen as this work was mainly built around Rojas-Méndez and Chapa (2019).

Reviewer 3 Report

Thank you for giving me the opportunity to review the article “Xenocentrism and consumer buying behavior: a comparative analysis of Malaysian vs Nigerian consumers”. Below my remarks:

  • the Authors have chosen an interesting and current topic on xenocentrism and consumer buying behavior;
  • “this study analyzed the direct effects of xenocentrism on the buying behavior of consumers towards imported products with perceived product quality, purchase intention, and product attitude as the variables used to measure consumers’ buying behavior” - despite this reference to the aim of the article - I suggest in the abstract and in the introduction to formulate a clear aim of the article;
  • In the final part of the Introduction, please describe synthetically what the individual sections of the article contain;
  • the literature review is correct;
  • section 3. RESEARCH OBJECTIVE, METHDOLOGY AND DATA should begin with a text, not a figure
  • the research is well designed and well described. The authors obtained interesting results;
  • the conclusions of the research are well described;
  • technical corrections are necessary. References should be properly prepared. E.g.
  • (Aranceta-Bartrina 1999a) Aranceta-Bartrina, Javier. Title of the cited article. Journal Title 6: 100–10.
  • (Aranceta-Bartrina 1999b) Aranceta-Bartrina, Javier. Title of the chapter. In Book Title, 2nd ed. Edited by Editor 1 and Editor 2. Publication place: Publisher, vol. 3, pp. 54–96.

Author Response

Thank you for giving me the opportunity to review the article “Xenocentrism and consumer buying behavior: a comparative analysis of Malaysian vs Nigerian consumers”. Below my remarks:

  • the Authors have chosen an interesting and current topic on xenocentrism and consumer buying behavior;
  • “this study analyzed the direct effects of xenocentrism on the buying behavior of consumers towards imported products with perceived product quality, purchase intention, and product attitude as the variables used to measure consumers’ buying behavior” - despite this reference to the aim of the article - I suggest in the abstract and in the introduction to formulate a clear aim of the article;

Done – based on comments from reviewer 1 and 2

  • In the final part of the Introduction, please describe synthetically what the individual sections of the article contain;

Done -

  • the literature review is correct;
  • section 3. RESEARCH OBJECTIVE, METHDOLOGY AND DATA should begin with a text, not a figure

Done

  • the research is well designed and well described. The authors obtained interesting results;
  • the conclusions of the research are well described;
  • technical corrections are necessary. References should be properly prepared. E.g.
  • (Aranceta-Bartrina 1999a) Aranceta-Bartrina, Javier. Title of the cited article. Journal Title 6: 100–10. (Aranceta-Bartrina 1999b) Aranceta-Bartrina, Javier. Title of the chapter. In Book Title, 2nd ed. Edited by Editor 1 and Editor 2. Publication place: Publisher, vol. 3, pp. 54–96.

Done

Round 2

Reviewer 1 Report

Dear Authors, 

Thank you for the improvements. The current version of the paper seems to be suitable for publishing. 

Reviewer 2 Report

Congratulations. The article is ready to be published.